# Performance of Ultra-Rapid Idylla™ EGFR Mutation Test in Non-Small-Cell Lung Cancer and Its Potential at Clinical Molecular Screening

**DOI:** 10.3390/cancers15092648

**Published:** 2023-05-07

**Authors:** Kenichi Suda, Kazuko Sakai, Tatsuo Ohira, Takaaki Chikugo, Takao Satou, Jun Matsubayashi, Toshitaka Nagao, Norihiko Ikeda, Yasuhiro Tsutani, Tetsuya Mitsudomi, Kazuto Nishio

**Affiliations:** 1Division of Thoracic Surgery, Department of Surgery, Kindai University Faculty of Medicine, Osakasayama 589-8511, Japan; 2Department of Genome Biology, Kindai University Faculty of Medicine, Osakasayama 589-8511, Japan; 3Department of Surgery, Tokyo Medical University, Shinjuku-ku, Tokyo 160-0023, Japan; 4Department of Diagnostic Pathology, Kindai University Hospital, Osakasayama 589-8511, Japan; 5Department of Anatomic Pathology, Tokyo Medical University, Shinjuku-ku, Tokyo 160-0023, Japan

**Keywords:** epidermal growth factor receptor (EGFR) mutation, molecular targeted therapy, companion diagnostics, personalized medicine

## Abstract

**Simple Summary:**

Genetic information is essential before starting the treatment of advanced-stage non-small-cell lung cancer (NSCLC) and in the adjuvant setting (*EGFR* mutation status only) after pulmonary resection of early-stage NSCLC. Several genetic tests for NSCLC are available, which vary in turnaround time and cost (usually higher costs for multi-gene tests). The Idylla™ EGFR Mutation Test is an ultra-rapid single-gene test used to detect *EGFR* mutations. In this study, we compared the performance of the Idylla EGFR Mutation Test with the current standard *EGFR* single-gene test (Cobas^®^ EGFR Mutation Test v2) and demonstrate the accuracy of the Idylla EGFR Mutation Test as a molecular screening platform. From these data, we propose genetic testing strategies that may reduce the costs and shorten the turnaround time in advanced-stage NSCLC and in the adjuvant setting after pulmonary resection of early-stage NSCLC.

**Abstract:**

Background: The Idylla™ EGFR Mutation Test is an ultra-rapid single-gene test that detects epidermal growth factor receptor (*EGFR*) mutations using formalin-fixed paraffin-embedded specimens. Here, we compared the performance of the Idylla EGFR Mutation Test with the Cobas^®^ EGFR Mutation Test v2. Methods: Surgically resected NSCLC specimens obtained at two Japanese institutions (N = 170) were examined. The Idylla EGFR Mutation Test and the Cobas EGFR Mutation Test v2 were performed independently and the results were compared. For discordant cases, the Ion AmpliSeq Colon and Lung Cancer Research Panel V2 was performed. Results: After the exclusion of five inadequate/invalid samples, 165 cases were evaluated. *EGFR* mutation analysis revealed 52 were positive and 107 were negative for *EGFR* mutation in both assays (overall concordance rate: 96.4%). Analyses of the six discordant cases revealed that the Idylla EGFR Mutation Test was correct in four and the Cobas EGFR Mutation Test v2 was correct in two. In a trial calculation, the combination of the Idylla EGFR Mutation Test followed by a multi-gene panel test will reduce molecular screening expenses if applied to a cohort with *EGFR* mutation frequency >17.9%. Conclusions: We demonstrated the accuracy and potential clinical utility of the Idylla EGFR Mutation Test as a molecular screening platform in terms of turnaround time and molecular testing cost if applied to a cohort with a high *EGFR* mutation incidence (>17.9%).

## 1. Introduction

Lung cancer is the leading cause of cancer-related mortality in the world. Driver mutation testing is an essential element of diagnostic procedures for advanced-stage non-small cell lung cancer (NSCLC) patients [1] and some surgically resected NSCLC patients. Multiplex genetic tests, such as the Oncomine Dx Target Test (Thermo Fisher Scientific, Wilmington, DE, USA), meet the latest recommendations for NSCLC screening in the advanced-stage setting, because multiplex analyses are time-saving compared with performing a series of single gene analyses [2,3]. However, it is also true that multiplex genetic tests are usually expensive. Because the reduction of medical costs is an urgent economic issues in many countries and as NSCLC is one of the most frequent malignancies in many countries [4], a strategy to reduce the cost of molecular testing for NSCLCs [5] without extending the turnaround time (TAT) is required. In addition, in the setting of surgically resectable NSCLC, *EGFR* mutation analysis is the only clinically required genetic test, because osimertinib, an EGFR inhibitor, is the only approved molecular targeted drug in the adjuvant setting in many countries including Japan for surgically resected pathological-stage (pStage) IB-III NSCLC patients with activating *EGFR* mutations [6].

The Idylla™ system (Biocartis, Mechelen, Belgium) is a simple, fully automated, real-time PCR (qPCR)-based platform that uses unextracted formalin-fixed, paraffin-embedded (FFPE) tissue sections as input material [7,8]. The Idylla EGFR Mutation Test has some advantages over other *EGFR* single gene tests, including an ultrarapid TAT (150 min for a single *EGFR* test), no requirement of specific technical skill, and a closed cartridge during the entire workflow (reducing the risk of contamination). Therefore, the Idylla system can be used in most laboratories with minimal infrastructure. These advantages of the Idylla system have accelerated the development of several other molecular testing platforms, including the Idylla GeneFusion assay, which assesses gene fusions in *ALK*, *ROS1*, *RET*, *MET* exon 14 skipping, and *NTRK*1/2/3, the Idylla BRAF Mutation Test, the Idylla KRAS Mutation Test, and the Idylla NRAS-BRAF Mutation Test. Several groups have performed comparison studies between the novel Idylla assays and conventional, clinically approved molecular testing platforms.

In this study, we performed a comparison of the Idylla EGFR mutation test with the Cobas^®^ EGFR Mutation Test v2 (Roche Diagnostics, Basel, Switzerland) in terms of the accuracy. We evaluated the performance of the Idylla EGFR Mutation Test and the Cobas EGFR Mutation Test v2 using FFPE specimens obtained from NSCLC patients who underwent surgical resection at two institutions in Japan. We also discuss a molecular testing strategy that incorporates the Idylla EGFR Mutation Test with the aim of reducing the molecular screening cost without extending the TAT.

## 2. Materials and Methods

### 2.1. Patients

A series of FFPE specimens were obtained from NSCLC patients who underwent surgical resection at the Kindai University Hospital (N = 100) or Tokyo Medical University Hospital (N = 70). After exclusion of four cases with scant/no tumor cells in the specimens (Figure 1), a total of 166 specimens were initially registered for the study. Patient characteristics are summarized in Table 1. The cohort consisted of 122 lung adenocarcinomas (73%), 6 adenosquamous carcinomas (4%), 35 squamous cell carcinomas (21%), and 3 pleomorphic carcinomas (2%). Following the 8th edition of the TNM classification, there were 100 pStage IA (60%), 20 pStage IB (12%), 37 pStage II–III (22%), and 9 pStage IV (5%) patients.

Serial FFPE sections from FFPE specimens used for DNA extraction were stained with hematoxylin and eosin to assess the proportion of tumor cells. Macro-dissection was used on 85% of specimens to enrich tumor cells (median proportion of tumor cells was 25%, range 20–70%). This study was approved by the ethical committees of the Kindai University Faculty of Medicine and the Tokyo Medical University Hospital (R03-191 and T2021-0281, respectively). Written informed consent was waived because of the retrospective nature of this study.

### 2.2. Study Design

The unstained FFPE specimens from the 166 NSCLC patients were analyzed by the Idylla EGFR Mutation Test at the Department of Genome Biology, Kindai University Faculty of Medicine. The Idylla EGFR Mutation Test is a single-use cartridge-based test designed for the qualitative detection of 39 different *EGFR* mutations. In this study, the Idylla investigator-use-only (IUO) assay was used.

The Cobas EGFR Mutation Test v2 was performed independently at LSI Medience Corporation (Tokyo, Japan) and SRL, Inc. (Tokyo, Japan). The Cobas EGFR Mutation Test v2 was used as the control because it is widely used in clinical practice and has shown a high concordance rate and ĸ value (97.5% and 0.938, respectively) compared with an NGS-based multiplex genetic test (Oncomine Dx Target test) [9].

For discordant cases between the Idylla vs. Cobas assays, the sample was reanalyzed with the Idylla EGFR Mutation Test. For discordant cases after the second Idylla EGFR Mutation Test, the Ion AmpliSeq Colon and Lung Cancer Research Panel V2 (CLV2, Thermo Fisher Scientific), which includes a single primer pool to amplify hotspots and targeted regions of 22 cancer genes frequently mutated in colorectal cancers and NSCLCs [10], was performed.

### 2.3. DNA Extraction and NGS-Based Panel Test

The Ion AmpliSeq Colon and Lung Cancer Research Panel V2 (CLV2) was performed as an NGS-based panel test. Briefly, DNA was isolated from FFPE specimens with the AllPrep DNA/RNA FFPE Kit (Qiagen, Venlo, Netherlands). The quality and quantity of the nucleic acid were verified with a NanoDrop 2000 device and PicoGreen dsDNA Reagent (both from Thermo Fisher Scientific). For library preparation, DNA was subjected to multiplex PCR amplification using the Ion AmpliSeq Library Kit 2.0 (Thermo Fisher Scientific) following the manufacturer’s protocol. Pooled libraries were subjected to the Ion Chef System (Thermo Fisher Scientific) for template preparation. Libraries were then loaded onto an Ion 550 chip and sequenced with the Ion S5 sequencing system. DNA sequencing data were accessed through the Torrent Suite version 5.12 program (Thermo Fisher Scientific). Reads were aligned with the hg19 human reference genome, and potential mutations were identified using Variant Caller version 5.12. Raw variant calls were filtered with a quality score of <100 and depth of coverage of <19 and were manually checked using the integrative genomics viewer (IGV; Broad Institute, Cambridge, MA, USA).

### 2.4. Statistical Analysis

Kappa statistics was used to compare the results of the Idylla EGFR Mutation Test with the Cobas EGFR Mutation Test v2. Statistical analyses were performed using GraphPad Prism software (Version 8; GraphPad Software Inc., La Jolla, CA, USA).

### 2.5. Total Expense Calculation for Molecular Testing

The total costs for molecular testing were calculated on the basis of medical insurance scores designated by the Japanese Pharmaceuticals and Medical Devices Agency (PMDA) in March 2023. The amount that medical institutions can claim for the molecular testing was determined by the PMDA for each examination. The total cost of a potential testing strategy that incorporates the Idylla EGFR mutation test was calculated on the basis of frequencies of *EGFR* mutations, as the cost of a single-plex EGFR test for patients with *EGFR* mutation and the sum of the costs of a single-plex EGFR test plus a following multiplex genetic test for patients without *EGFR* mutation. We calculated the *EGFR* mutation frequency at which the cost of molecular testing that incorporates the Idylla EGFR mutation test becomes less than the cost of molecular testing with a multiplex genetic test for all patients.

## 3. Results

### 3.1. Performance of the Idylla EGFR Mutation Test

As shown in Figure 1, both the Idylla and the Cobas assays were successfully performed in 165 cases out of 166 (success rates: 99.4%). Because one specimen was invalid for both assays, the quality of the sample was assumed to be low. The detailed results of the 165 specimens are summarized in Table 2. Among the 165 samples, 52 were positive for *EGFR* mutation (21 samples had exon 19 deletion and 28 had L858R, 2 had G719X, and 1 sample had L861Q point mutations) and 107 were negative for *EGFR* mutation in both assays (overall concordance rate: 96.4%). The ĸ value between the two assays was 0.920 (95% confidential interval: 0.857–0.983). Discordant results were seen in six cases (Table 2).

The six cases were subjected to a second Idylla assay. One case (exon 19 deletion by the Cobas assay but a negative result in the initial Idylla assay) showed exon 19 deletion in the second Idylla assay. The other five cases showed the same results at the second Idylla EGFR Mutation Test. An NGS-based panel test was performed on the five discordant cases between the Idylla vs. Cobas assays.

### 3.2. Detailed Examination of the Discordant Cases

An NGS-based panel test (CLv2) was performed for the five discordant samples. In the cases with positive results by the Idylla assay (L858R or exon 20 insertion) but negative results by the Cobas assay, the NGS results were concordant with the Idylla assay. The detected exon 20 insertion was Asn771_Pro772insThr, which was not covered by the Cobas assay. The reason why the Cobas assay could not detect L858R *EGFR* point mutation is not clear. The NGS-based panel test revealed that the specimen had compound *EGFR* mutations (L858R plus L776H). The L776H *EGFR* mutation is not covered by the Idylla assay or Cobas assay, which may clarify why the sample was determined as L858R point mutation by the Idylla assay.

Three cases had positive results shown by the Cobas assay (all showed exon 19 deletion) but negative results by the Idylla assay. In the NGS-based analysis, one had a rare exon 19 deletion (Leu747_Lys754delinsSerThr) that is not covered by the Idylla assay, one had an *EGFR* exon 19 insertion mutation (Lys745_Glu746insIleProValAlaIleLys), and the other had *EGFR* L747P point mutation of the exon 19. Therefore, we concluded that the Idylla assay was correct in two cases and the Cobas assay was correct in one case.

Because primer sequences of the Cobas EGFR Mutation Test v2 are not available, it was not possible to elucidate the reason for the false positive results. However, we noticed a shared sequence (… AAG GAA CCA…) between our case with the *EGFR* L747P point mutation (K-38) and one of the detectable exon 19 deletion mutations (Leu747_Thr751insPro) by the Cobas assay (Figure 2). Because the clinical genetic test for this patient (K-38), which was performed independently using the Cobas EGFR Mutation Test v2, also detected the exon 19 deletion, it is possible that the Cobas assay may call a false positive result for this rare *EGFR* exon 19 point mutation (L747P). However, this phenomenon should be confirmed in future studies using other NSCLC specimens with this rare *EGFR* mutation. The reason for the false-positive results by the Cobas EGFR Mutation Test v2 in the specimen with *EGFR* exon 19 insertion mutation (Lys745_Glu746insIleProValAlaIleLys) is unclear.

## 4. Discussion

In this study, we found that the performance of the Idylla EGFR Mutation Test is comparable with that of the Cobas EGFR Mutation Test v2. The overall concordance rate of both assays was 96.4% and the ĸ value between the two assays was 0.920. This concordance rate was comparable with the result of a recently reported FACILITATE study, a real-world, prospective, multicenter European study that evaluated the performance of the Idylla EGFR Mutation Test with local reference methods (either NGS-based, Cobas, Therascreen EGFR RGQ PCR, Sanger sequencing, or other methods) in 16 sites [11]. In the FACILITATE study, the overall percentage agreement between the Idylla assay and local reference methods was 97.7%; the positive agreement was 87.4%, the negative agreement was 99.2%, and there were 38 (2.6%) discordant cases. In the analysis of discordant cases using a third method analysis such as digital droplet PCR (ddPCR), the mutant allele frequencies were quite low (0.4–4%) in cases with discordant-negative cases for the Idylla assay. In a third method analysis for discordant-positive cases, the Idylla assay was correct in three cases and the local reference method was correct in five cases. In our study, we compared PCR-based *EGFR* single gene analyses (Idylla EGFR Mutation Test vs. Cobas EGFR Mutation Test v2); after re-testing some discordant cases, the Idylla assay was found to achieve the correct results (indicating the Cobas assay had false negative or false positive results) in a few specimens. These data, together with the results of the FACILITATE study, support the clinical application of the Idylla EGFR Mutation Test as a single-plex *EGFR* mutation test for NSCLC patients.

The most Important advantage of the Idylla EGFR Mutation Test is the ultra-rapid TAT (~150 min). We thus consider how to apply this test in the era of multiplex genetic tests (Figure 3A). Because the *EGFR* mutation is one of the most frequent driver mutations in NSCLC, and as driver mutations in NSCLCs are usually present in mutually exclusive fashion, performing the Idylla EGFR Mutation Test prior to a multiplex genetic test may be a reasonable strategy to reduce the medical expenses of molecular profiling for NSCLC patients (Figure 3B). In fact, several previous studies reported the feasibility of performing the Idylla EGFR Mutation Test prior to NGS-based assays [12,13], although these studies focused on prompt *EGFR* mutation testing for patients under oncological emergency. 

Here we calculate the total expenses for molecular profiling using the medical insurance scores designated by the Japanese PMDA in March 2023; the insurance score for a single-plex EGFR test is 2500 (=25,000 yen) and the insurance score for a multiplex genetic test is 14,000 (=140,000 yen) for the Oncomine Dx Target Test. As shown in Figure 3C, the total estimated cost will become smaller in the combined Idylla assay plus multiplex test (Figure 3B) compared with the use of multiplex test only (Figure 3A), if applied for a cohort with an *EGFR* mutation frequency higher than 17.9%. Therefore, the combination test will reduce the total cost of molecular profiling, at least in Japan, if used for lung adenocarcinoma patients, in which the frequency of the *EGFR* mutation is approximately 30% or higher [14]. A recent case study reported that the Idylla EGFR Mutation Test failed to detect a rare *EGFR* exon 19 deletion (*EGFR* L747_A755delinsSS), which is not covered by the Idylla EGFR Mutation Test [15]. This is in line with the application of Idylla cartridges, which are intended to identify common, clinically relevant Tier 1 and 2 mutations and not rare or complex variants [16]. In our proposed molecular testing strategy, patients with a potentially targetable rare *EGFR* mutation will be identified by a subsequent NGS-based multiplex genetic testing such as the Oncomine Dx Target Test (Figure 3B). 

In the early-stage setting after curative pulmonary resection, the current essential biomarker testing involves *EGFR* mutation analysis and PD-L1 staining to decide the adaptation of adjuvant treatment using osimertinib [6] or atezolizumab [17]. Because ASCO guideline updates [18], and ESMO consensus statements [19] indicate that adjuvant osimertinib is recommended for NSCLC patients with an *EGFR* sensitizing mutation regardless of the PD-L1 expression status, the ultra-rapid Idylla EGFR Mutation Test will provide a potential to avoid unessential PD-L1 testing if the examined specimen had an *EGFR*-sensitizing mutation.

Notably, PCR-based single-plex *EGFR* mutation analyses, including the Idylla and Cobas assays, have a disadvantage of *EGFR* mutation coverages (87–98%) compared with NGS [12,13,20]. These data support the usefulness of comprehensive genomic profiling during the treatment course of NSCLC patients [21]. In this analysis, we incidentally observed that the Cobas assay reported false-positive results (calling *EGFR* exon 19 deletion in specimens with *EGFR* exon 19 L747P point mutation or exon 19 insertion). The detailed mechanisms and the incidence of such false positive results is unclear because of the lack of sufficient data [22]. Clinicians should be aware that patients may lose the opportunity to receive adequate treatment because of false-positive or false-negative biomarker testing results.

*EGFR* L747P point mutation, which results from a double thymine-to-cytosine (TT > CC) transition (Figure 2), comprises less than 1% of *EGFR* mutations [23]. The current standard of care for NSCLC patients with *EGFR* exon 19 deletion is osimertinib [24]. However, a recent structure-function analysis of various *EGFR* mutation variants [25] indicated that the L747P mutation is classified into “P-loop and C-helix compressing group” that will show inherent resistance to 1st- or 3rd-generation TKIs but is sensitive to 2nd-generation TKIs such as afatinib or dacomitinib. Several case studies have reported on the efficacy of afatinib or dacomitinib and low efficacy of gefitinib or osimertinib in NSCLC patients with *EGFR* L747P point mutation [23,26,27]. 

Another *EGFR* mutation that led to a false-positive result in the Cobas assay was a rare *EGFR* exon 19 insertion mutation (Leu747_Lys754delinsSerThr). Several studies have found that tumors with *EGFR* exon 19 insertion mutation are usually sensitive to EGFR-TKIs [28,29], and therefore these mutations should be included in the list of detectable driver mutations in the near future. Additionally, compound *EGFR* mutations, which are often missed through the use of mutation-specific assays, may lead to varying responses to EGFR-TKIs. These observations also highlight the importance of comprehensive genomic profiling in NSCLC patients at least once during the treatment course.

On the basis of several advantages of the Idylla assays, other molecular testing platforms including Idylla GeneFusion assay, the Idylla BRAF Mutation Test, the Idylla KRAS Mutation Test, and the Idylla NRAS-BRAF Mutation Test have been developed. In a comparison analysis of the Idylla GeneFusion assay for previously characterized fusion-positive tumors (37 NSCLCs and 2 parotid gland carcinomas), the Idylla GeneFusion assay successfully detected 36 fusions (overall agreement: 92.3%) [30]. Another group compared the performance of two ultrafast gene fusion assays (the Idylla GeneFusion assay and the NGS-based Genexus assay) in 195 NSCLC cases (113 known gene fusions and 82 wild-type tumors). The accuracy was 92.3% and 93.1% for the Idylla assay and the Genexus assay, respectively [31].

Comparisons of the Idylla assay with other genetic aberrations have been performed in solid tumors other than NSCLCs. For example, in the analysis of *KRAS*, *NRAS*, and *BRAF* mutations in 850 colorectal cancer (CRC) cases, the concordance rate was 88.6%, accounting for all mutations including those not in the Idylla cartridge by design [32]. The colorectal team in our department also performed a comparison study between the Idylla assay and the MEBGEN RASKET-B assay for *KRAS*, *NRAS*, and *BRAF* mutations using 253 CRC specimens. The authors observed the potential clinical usefulness of the Idylla assay, showing a high concordance rate of 97.4%, a negative concordance rate of 95.7%, and overall concordance rate of 95.3% (κ = 0.919, 95% CI 0.871–0.967) [33]. Another group compared the Idylla assay with anti-BRAF V600E (clone VE1) immunohistochemistry in 90 melanoma samples. The agreement rate of the assays was 91% (72/79) [34]. Thus, these comparison studies, including the current study, have revealed the accuracy and clinical usefulness of various Idylla assays.

## 5. Conclusions

We demonstrated the comparability of the ultra-rapid Idylla EGFR Mutation Test compared with the Cobas EGFR mutation test v2. The Idylla EGFR Mutation Test may have the potential to reduce the cost of driver mutation testing if used in a cohort with high *EGFR* mutation rates without extending the TAT; furthermore, it may dramatically reduce the TAT if patients have *EGFR* mutation. We also observed the limitations of the PCR-based *EGFR* genetic tests in terms of false-positive results (especially for the Cobas EGFR mutation test v2) and false-negative results, including the detection of the *EGFR* compound mutations. While these phenomena might be rare, clinicians should keep the possibility of false-positive/negative results in mind when treating NSCLC patients.

## Figures and Tables

**Figure 1 cancers-15-02648-f001:**
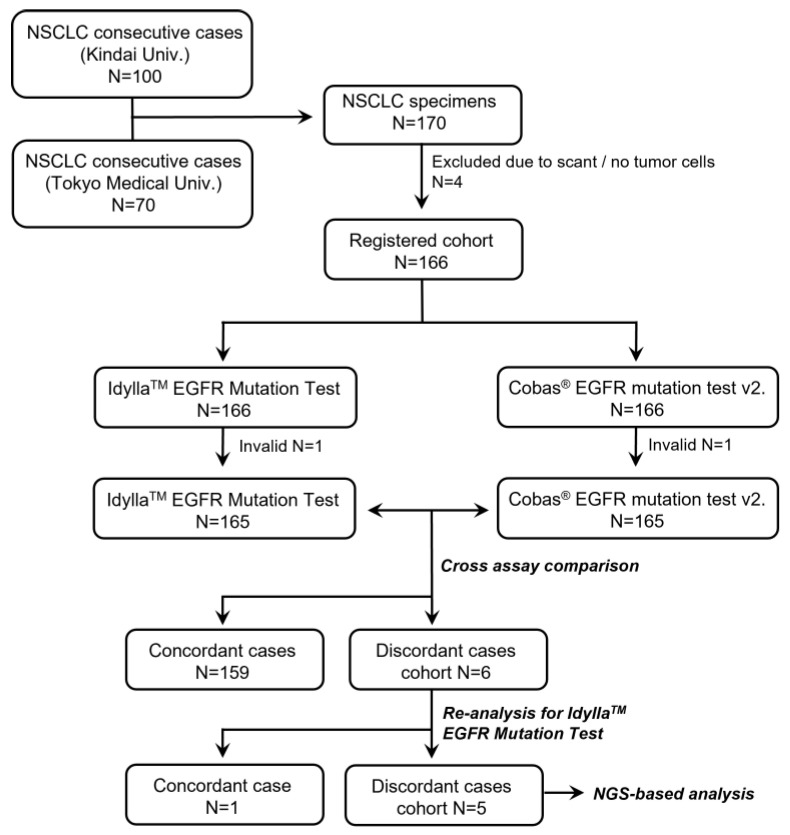
Flowchart of the study.

**Figure 2 cancers-15-02648-f002:**
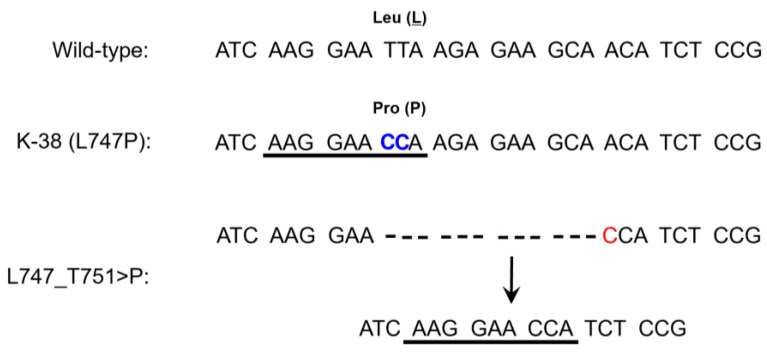
Hypothesis for a false-positive result by the Cobas EGFR mutation test v2 in a tumor with *EGFR* L747P point mutation (K-38). Similarity of sequences between L747P point mutation and a detectable *EGFR* exon 19 deletion (L747_T751>P) by the Cobas EGFR mutation test v2 were highlighted by the underline (AAG GAA CCA). The mutated nucleotides are highlighted in color including the double thymine-to-cytosine (TT > CC) transition in L747P mutation.

**Figure 3 cancers-15-02648-f003:**
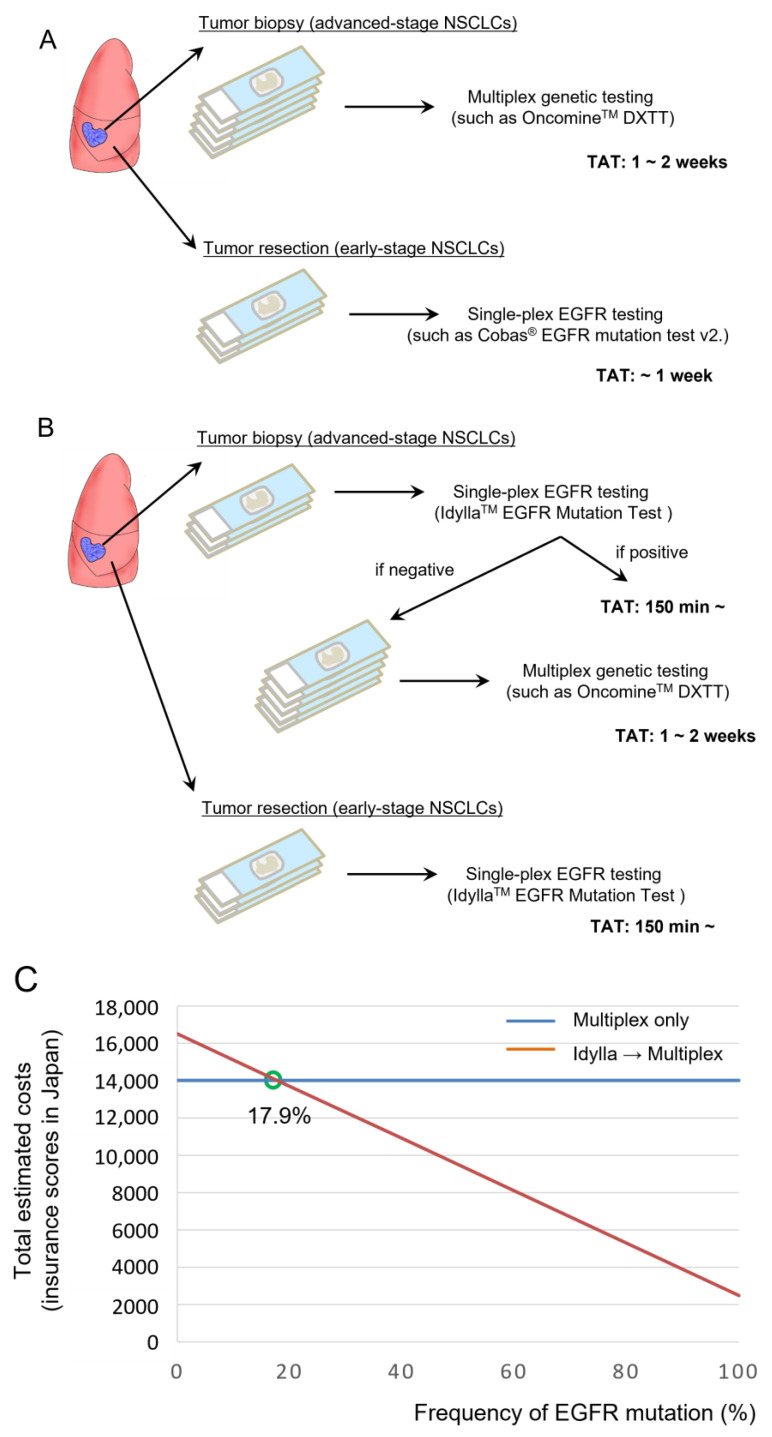
Incorporation of the Idylla EGFR mutation test before multiplex genetic testing in a cohort with high *EGFR* mutation frequency. (**A**) Current standard of care for molecular profiling of advanced-stage NSCLCs and early-stage NSCLCs. (**B**) Potential strategies to incorporate the Idylla EGFR mutation test. (**C**) Total estimated costs of multiplex testing vs. the Idylla EGFR mutation test followed by multiplex testing (if *EGFR* is negative) in cohorts with various *EGFR* mutation frequencies.

**Table 1 cancers-15-02648-t001:** Clinical and pathological characteristics of patients.

Factors		Registered Cases (N = 166)
Sex	Female	56 (34%)
	Male	110 (66%)
Age	Median (range)	72-year-old (35–90)
Histology	Adenocarcinoma	122 (73%)
	Adenosquamous carcinoma	6 (4%)
	Squamous cell carcinoma	35 (21%)
	Pleomorphic carcinoma	3 (2%)
pStage (8th Edition)	IA1-3	100 (60%)
	IB	20 (12%)
	II-III	37 (22%)
	IV	9 (5%)
Macro-dissection	Yes	141 (85%)
	No	25 (15%)
Proportion of tumor cells	Median (range)	25% (20–70)

**Table 2 cancers-15-02648-t002:** Concordance summary in all patients (N = 165) with valid results for *EGFR* testing.

Detected Mutation		Idylla					
		Exon 19 del	L858R	G719X	Exon 20 ins	L861Q	Wild-Type
Cobas	Exon 19 del	21 *	-	-	-	-	4 **
	L858R	-	28	-	-	-	-
	G719X	-	-	2	-	-	-
	Exon 20 ins	-	-	-	-	-	-
	L861Q	-	-	-	-	1	-
	Wild-type	-	1 ^#^	-	1	-	107

* Concurrent T790M was detected in one specimen for both assays. ** One case turned out to be exon 19 del at the second Idylla EGFR Mutation Test. ** The other three had a rare exon 19 del (Leu747_Lys754delinsSerThr), exon 19 insertion, and L747P mutation. ^#^ L858R + L776H were detected by NGS-based analysis.

## Data Availability

The data presented in this study are available on request from the corresponding author.

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
