# Peer review of "Performance of Ultra-Rapid Idylla™ EGFR Mutation Test in Non-Small-Cell Lung Cancer and Its Potential at Clinical Molecular Screening"

_cancers, 2023, doi:10.3390/cancers15092648_

Round 1
Reviewer 1 Report
The authors have compared the performance of the Idylla EGFR Mutation Test with that of the standard single-gene Cobas EGFR Mutation Test v2 in terms of the accuracy, turnaround time (TAT) and cost. They determined that the Idylla EGFR Mutation Test may reduce the cost and shorten the TAT in patients with non-small cell lung cancer.
This study has brought to light interesting findings regarding the accuracy, cost and TAT of the ultra-rapid single-gene test, Idylla EGFR Mutation Test. However, some pertinent information is lacking.
Comments:
1. The authors performed an NGS-based panel test (CLv2) for analyzing five discordant samples. However, they have not described the method. Please add this information in the Methods section.
2. In Fig. 3C, the authors have shown the total estimated cost of the Idylla EGFR Mutation Test and multiplex test. Please provide the basis of the calculation formula in the Methods section.
Author Response
We appreciate the time and effort you and the reviewers have dedicated to providing insightful feedback on ways to strengthen our paper. We have modified our manuscript following the suggestions raised by the reviewer. We also split our Figure 2 into a Table (Table 2) and a Figure (revised Figure 2) following your comment in the manuscript. To facilitate your review of our revision, the following is a point-by-point response to the comments. Comment 1: The authors performed an NGS-based panel test (CLv2) for analyzing five discordant samples. However, they have not described the method. Please add this information in the Methods section. Response 1: Thank you for your comments and suggestions. Following the comment by the reviewer, we have added the methods of the NGS-based panel test (CLv2) in the Methods section (Page 5, Lines 132-147). “2.3. DNA extraction and NGS-based panel test The Ion AmpliSeq Colon and Lung Cancer Research Panel V2 (CLV2) was performed as an NGS-based panel test. Briefly, DNA was isolated from FFPE specimens with the AllPrep DNA/RNA FFPE Kit (Qiagen, Venlo Netherlands). The quality and quantity of the nucleic acid were verified with a NanoDrop 2000 device and PicoGreen dsDNA Reagent (both from Thermo Fisher Scientific). For library preparation, DNA was subjected to multiplex PCR amplification using the Ion AmpliSeq Library Kit 2.0 (Thermo Fisher Scientific) following the manufacturer’s protocol. Pooled libraries were subjected to the Ion Chef System (Thermo Fisher Scientific) for template preparation. Libraries were then loaded onto an Ion 550 chip and sequenced with the Ion S5 sequencing system. DNA sequencing data were accessed through the Torrent Suite version 5.12 program (Thermo Fisher Scientific). Reads were aligned with the hg19 human reference genome, and potential mutations were identified using Variant Caller version 5.12. Raw variant calls were filtered with a quality score ofReviewer 2 Report
Authors have reported the Performance of Ultra-Rapid IdyllaTM EGFR Mutation Test in 2 Non-Small Cell Lung Cancer and Its Potential at Clinical 3 Molecular Screening but the following should be considered
1. simple summary is not needed it must be included in the abstract.
2. abstract must give the important findings
3. Rationale of the work is needed in the introduction
4. In the introduction epidemelogy of lung cancer need to be given
5. Result needs to be discussed more
6. Check for spelling and english errors
7. References need to be rechecked.
Author Response
We appreciate the time and effort you and the reviewers have dedicated to providing insightful feedback on ways to strengthen our paper. We have modified our manuscript following the suggestions raised by the reviewer. We also split our Figure 2 into a Table (Table 2) and a Figure (revised Figure 2) following your comment in the manuscript. To facilitate your review of our revision, the following is a point-by-point response to the comments.
Comment 1: simple summary is not needed it must be included in the abstract.
Answer: Thank you for your review and for your valuable comments. Many recent original papers published in Cancers have Simple Summary in addition to the Abstract. Therefore, we would like to keep it in the revised manuscript. However, we are happy to delete the Simple Summary if requested by the Editorial Office at the time of proof-reading.
Comment 2. abstract must give the important findings
Answer: We modified the conclusion of the abstract so that it gives the important findings of our study (Page 1, Lines 42-44).
“We demonstrated the accuracy and potential clinical utility of the Idylla EGFR Mutation Test as a molecular screening platform in terms of turnaround time and molecular testing cost if applied to a cohort with a high EGFR mutation incidence (> 17.9%).”
Comment 3: Rationale of the work is needed in the introduction.
Response 3: We added our hypothesis to describe the rationale of the work in the Introduction (Page 3, Lines 85-92).
“In this study, we performed a comparison of the Idylla EGFR mutation test with the Cobas® EGFR Mutation Test v2 (Roche Diagnostics, Basel, Switzerland) in terms of TATs and accuracy. We evaluated the performance of the Idylla EGFR Mutation Test and the Cobas EGFR Mutation Test v2 using FFPE specimens obtained from NSCLC patients who underwent surgical resection at two institutions in Japan. We also discuss a molecular testing strategy that incorporates the Idylla EGFR Mutation Test with the aim to reduce the molecular screening cost but not extend the TAT.”
Comment 4: In the introduction epidemiology of lung cancer need to be given.
Response 4: Following the comment raised by the reviewer, a short sentence of the epidemiology of lung cancer was added in the Introduction (Page 3, Line 52).
“Lung cancer is the leading cause of cancer-related mortality in the world.”
Comment 5: Result needs to be discussed more.
Response 5: We have discussed all findings of our manuscript, including the accuracy of the Idylla EGFR mutation test (including previous data), potential mechanisms of false positive/negative results observed in this study either by the Cobas assay or the IdyllaTM assay, and potential as an useful clinical tool as a molecular testing platform.
Comment 6: Check for spelling and english errors
Response 6: Native English check was performed during this revision process. Dr. Gabrielle White, from Edanz, have checked the entire manuscript and have made minor modifications (all of these English checks were also highlighted). In the Acknowledgments section, a short sentence “We thank Gabrielle White Wolf, PhD, from Edanz (https://jp.edanz.com/ac) for editing a draft of this manuscript.” was added.
Comment 7: References need to be rechecked.
Response 7: We checked our reference list and confirm that the reference list is correct. Along with the revision of the manuscript, the order of the reference 4 and 5 has been switched.
Round 2
Reviewer 2 Report
Authors adequately answered on comments.